# User Innovativeness and Fintech Adoption in Indonesia

**Budi Setiawan** [1,2,*] **, Deni Pandu Nugraha** [3]**, Atika Irawan** [4]**, Robert Jeyakumar Nathan** [5] **and Zeman Zoltan** [1]

1 Doctoral School of Economic and Regional Sciences, Hungarian University of Agriculture and Life Sciences, 2100 Godollo, Hungary; zeman.zoltan@uni-mate.hu
2 Faculty of Economics, Universitas Indo Global Mandiri, Palembang 30129, Indonesia
3 Department of Management, UIN Syarif Hidayatullah Jakarta, Tangerang Selatan 15412, Indonesia; denipandu.nugroho@uinjakarta.ac.id
4 School of Business and Management, Institut Teknologi Bandung, Bandung 40116, Indonesia; atika.irawan@sbm-itb.ac.id
5 Faculty of Business, Multimedia University, Cyberjaya 63100, Malaysia; robert.jeyakumar@mmu.edu.my
* Correspondence: setiawan.budi@hallgato.uni-szie.hu or budi.setiawan@uigm.ac.id

**Abstract:** The integration of the financial industry and financial technology (Fintech) plays a pivotal role in increasing financial services reach and inclusion for the large unbanked population in Indonesia. Fintech adoption optimization expands the financial access to formal financial institutions, especially to vulnerable groups such as the unbanked population who predominantly reside in rural areas far from formal financial institutions. Fintech is viewed as a game changer to bring finance to the unreached communities via information technology and digital financial landscape. In this causal research, data collection was done via online questionnaires to 485 Fintech users between December 2020 and April 2021. Data analysis and path modelling was performed using smartPLS 3.0 software. Result shows user innovativeness as a significant predictor, directly and indirectly affecting the adoption of Fintech in Indonesia, while user attitude found the most important factor towards Fintech adoption. Financial literacy is the least important variable to predict Fintech adoption, contrary to popular belief. This indicates that Fintech usage requires less financial literacy and is potential to reach unbanked population and those with low financial literacy. To make Fintech more inclusive, the government needs to accelerate improving Information and Communications Technology (ICT) infrastructure such as widening mobile broadband penetration and soft infrastructure by encouraging Fintech startup, allowing regulatory sandbox for startups, and driving financial institutions to innovate through Fintech to bring financial services to unbanked population.

**Keywords:** financial technology; attitude; financial literacy; individual innovativeness; unbanked population; Sustainable Development Goal 8

## 1. Introduction

Innovation, technology, and advancements in information and communication technologies have affected every facet of human life as it brings incremental changes to the economy. These advancements have brought about disruptive changes in the financial sector. The development of financial technology (Fintech) is an innovation that helps people do financial transactions with ease and speed. Fintech portrays wider user-reach offering financial services and its quickly gaining user bases across the globe. In adopting Fintech, customers are more influenced by the perception of benefit compared to the risk associated with Fintech adoption [1,2]. Indonesia, as a developing country, not only has limited funds to enhance its financial accessibility and infrastructure, the country also has high percentage of unbanked population. Fintech could serve as a game changer in bringing financial products to the currently unreached and unbanked population in Indonesia. Referring to World Bank Global Findex 2017, the unbankable population in Indonesia reaches 95 million population, behind Pakistan (100 million), India (190 million), and China (225 million) [3]. Fintech adoption can benefit the unbankable population,

particularly in peer-to-peer services for financing. This opportunity is supported by the population that uses mobile telecommunication in Indonesia. Davis et al. [4] stated that 85 percent of the population in Indonesia has a mobile phone but only 64.8 percent of the population actively use the internet. It shows that Indonesian society is open and ready to accept innovations. The adoption of Fintech in a country is believed to help the unmet demand and increase financial inclusivity of the society [5].

The advantages of Fintech adoption to communities is aligned with the Sustainable Development Goals (SDGs) and United Nation Development Program (UNDP) such as promoting zero poverty, ending hunger, or providing food security [6]. It also supports the agenda for clean energy adoption and preserves the ecosystem in a circular economy. However, to enhance its use, the customer must have a better understanding of the Fintech products and services by acknowledging the benefits and risks of Fintech adoption [1]. Willingness to try and use new financial services and products is not enough. Society must also have a culture for open innovation to accelerate Fintech adoption [7]. The recent pandemic of COVID-19 has shown that people able to tech-educated in a short time and force to manage their finance digitally.

Previous study has linked Fintech adoption with several theories, including the technology acceptance model (TAM), united theory of acceptance and use of technology (UTUAT), technology readiness (TR), theory of interpersonal behavior (TIB), institutional theory (IT), and individual innovativeness theory (IIT). Faraj and Pachidi [8] also found that institutional theory is beneficial in examining Fintech adoption. This study has chosen to focus on the TAM, IT, and IIT with some considerations; first, TAM is the leading theory to measure technology adoption [9] efficient in forecasting and explaining the new technology acceptance [10]. Second, IT is the foundation of an institutional ecosystem that can promote Fintech services and engage with technology [8]. Davis and Sinha [11] reveal that institutional factors play a pivotal role in reinforcing information and communication technology (ICT)'s innovations to reorganize the corporations, including Fintech companies. Finally, Rogers [12] describes IIT as levels of individual ability to adapt to a technology compared to others. Individual differences in the acceptance of a new change led to the initiation of individual innovativeness. Individual innovation plays a pivotal role in accepting new technology [13,14].

In Indonesia, Financial Service Authority supports Fintech adoption by its regulations. While it is more focused on Fintech in peer-to-peer services, the updated law regarding security and financial literacy of the society is expected to develop as the regulator gets more experience [4]. Fintech development cannot depend only on the regulation to grow, but it needs to consider its user perspective and knowledge. A study in China found that government support is essential to boost user innovation and enhancement Fintech adoption [15]. Hence, the government may use this research to create new regulations and support Fintech adoption in Indonesia. The novelty of this research use factors that influence Fintech adoption, such as brand image, financial health, and user innovativeness by including 485 respondents. At the time others were limited to financial literacy and government involvement.

The following section of this study, Section 2, presents theoretical background, previous studies, and the methodology. Section 3 describes the result and follow by its discussion in Section 4. Finally, Section 5 concludes the research with implication, recommendation, and limitation.

## 2. Literature Reviews, Proposes Hypotheses, and Methodology

### 2.1. Literature Review and Proposes Hypotheses

The connection between finance and technology has a long history. Arner, Barberis, and Buckle [16] divide the development of Fintech into three stages; first, Fintech 1.0 can be traced back around 1866 to 1967 when transatlantic telegraph cable was built to support the telecommunication system. The proliferation of digital technology and communication from 1967 to 2008 became the trigger for the birth of Fintech 2.0. Currently, the Fintech

sector has entered its third phase, reinforced by the presence of new start-ups that can interlink between financial products and technology to directly provide the needs of financial services for individuals and companies.

Nowadays, the existence of Fintech is more advanced than a decade ago. Fintech service is not only applied to the financial and banking industry, but also has expanded to various other sectors such as telecommunications, airlines, and wholesales [17]. The development of Fintech is driven by various advantages over conventional financial services, such as real-time process with 24 h and seven days service with a low cost, which has an excellent benefit for people, especially the rural society who lives far from the bank ecosystem [18]. The convenience of Fintech and its economic benefit encourage people to adopt and use it in daily life despite their concern to security risk that embed in the technology [19]. Government intervention accelerate the penetration of Fintech service such as e-payment [20].

Fintech service is still not adopting equally in many countries as describe by Ernst & Young Report [21] that 87 out of 100 consumers in China and India have adopted Fintech, compared to 35 and 34 in France and Japan, respectively. The study from Ryu [22] explains that the barrier factors of Fintech adoption include financial risk, legal uncertainty, security and privacy, and the inadequate operational systems of Fintech companies. By embracing innovation, people may alleviate the risk that appear from adopting new technology [23]. Furthermore, low levels of financial literacy are also challenging to public awareness in Fintech adoption [24,25]. The initiation to investigate the impact of technology adoption was conducted by Davis [26] by proposing TAM to test users' internal beliefs toward the acceptance of information technology. The TAM model aims to identify the modifications that must be implemented to the new technology accepted for users and be adopted in a final stage. Several recent works have integrated TAM and technology adoption in varying sectors; i.e., TAM to measure adoption in insurance customers' intentions [27], mobile banking [28], and credit card [24].

Hu et al. [29] extended the TAM model by adding new variables such as user innovation and government support as Fintech adoption determinants in China. The structural equation modeling (SEM) analysis revealed a significant correlation between brand image, government support and user innovation to Fintech adoption. In contrast, perceived ease of use does not significantly impact Fintech usage in China.

The recent study by To and Trinh [25] investigates Fintech adoption in Vietnam by focusing on the influence of behavioral intention (perceived ease of use, perceived usefulness), trust, and perceived enjoyment of mobile-wallets services. The result implies that perceived ease of use, perceived usefulness, and enjoyment have positive and significant impacts on behavioral intention to use M-wallets, while there is no direct impact from trust to M-wallets usage in Vietnam.

Further, the institutional theory (IT) explains the process of organizational and reasons behavior, including the effect of organizational patterns in a broader scope. Currently, IT has shifted from a closed system to an open system, including embracing various aspects, such as technology [30]. For example, Teigland et al. [31] depict that institutional theory integrates the institutional-level changes that occur, as has been the case with the emergence of Fintech. A new study integrating institutional theory and Fintech adoption is conducted by Braido and Klein [32] in Brazil. Their study concludes that institutional theory supported the organizational changes related to the mobile payment system's regulatory, normative, and cultural aspects.

### 2.2. The Determinants of Fintech Adoption

This study aims to investigate the determinants of Fintech adoption in Indonesia. Several variables such as financial health (FH), brand image (BI), perceived ease to use (PEU), perceived usefulness (PU), attitude (AT), financial literacy (FL), user innovativeness (UI), and government support (GS) is applied as an independent variable, which predicted to influence Fintech adoption in Indonesia. Referring to Hu et al. [29], these determinant

variables are adopted from TAM, the most widely used and good explanator for Fintech adoption. To extend the analysis, financial literacy, and financial health are added to predict Fintech adoption.

### 2.2.1. Financial Health (FH)

Joo [33] defines an individual's financial health as a comprehensive concept that cannot be identified by one measure since it integrates financial satisfaction, financial situation, attitudes, and behaviors. For this study's purposes, financial health is evaluated by looking at respondents' attitudes regarding financial allocation and management during the COVID-19 pandemic. Previous studies found that financial health has a positive relationship with Fintech adoption [25,34]. Based on previous research, the proposed hypothesis is:

**Hypothesis 1 (H1)**. *There is a positive relationship between financial health and Fintech adoption in Indonesia*.

### 2.2.2. Brand Image (BI)

Brand image plays an essential role in creating trust for financial technology users. Moreover, transactions of Fintech services are carried out without involving direct contact among parties. Previous research on brand image and technology adoption was conducted by analyzing various aspects, for example, the brand image associated with quality [35]; brand image is integrated with a brand equity [36]. This study looks at a brand image with user preferences in choosing Fintech companies based on well-known brands, including company reputation. The empirical study of Hu et al. [29] and Caviggioli et al. [37] explains that brand image has a positive relationship with Fintech adoption. Therefore, the proposed hypothesis is:

**Hypothesis 2 (H2)**. *There is a positive relationship between brand image and Fintech adoption in Indonesia.*

### 2.2.3. Perceived Ease to Use (PEU)

Perceived ease of use (PEU) is associated with the level of individual effort to use new technology [26]. In this study, PEU was measured by the efficiency when using Fintech services, including assessing the Fintech service interface, and the ease of access to Fintech services from various electronic devices. Previous research that integrated PEU with technology adoption was conducted by Kanchanatanee et al. [38]; Wang et al. [39], and Hu et al. [29]. All the results of these studies explain that PEU has a significant relationship with technology adoption, except Kanchanatanee et al. [38] reveals that there is no correlation between perceived ease of use and technology adoption but found there is an indirect relationship between perceive to use and fintech adoption. The study also analyzes the relationship between PEU and Fintech adoption with the perceived usefulness as a mediated variable by considering that perceived usefulness is often plays a pivotal role in technological adoption. Based on the empirical study above, the following hypothesis is proposed:

**Hypothesis 3a (H3a)**. *There is a positive relationship between perceived ease of use and Fintech adoption in Indonesia.*

**Hypothesis 3b (H3b)**. *There is a positive indirect relationship between perceived ease of use and Fintech adoption in Indonesia mediated by perceived usefulness.*

### 2.2.4. Perceived Usefulness (PU)

Davis et al. [26] explained the perceived usefulness (PU) of the extent to which the technology can help improve performance. This variable plays a pivotal role in predicting technology adoption [40]. PU in this study is measured by evaluating to what extent Fintech

adoption can fulfill the user needs, such as Fintech services can save time, and Fintech services provide benefits to users. Previous research has explained that PU positively correlates with technology adoption [41–43]. Based on previous research, the proposed hypothesis is:

**Hypothesis 4 (H4)**. *There is a positive relationship between perceived usefulness and Fintech adoption in Indonesia.*

### 2.2.5. Attitude (AT)

Ajzen [44] defines the attitude as a person's disposition to assess likes and dislikes towards an object, behavior, person, institution, or event. In this study, attitude is measured by investigating whether someone believes that using Fintech services is a good idea, their comfort level, and their interest in those services. Previous studies regarding attitudes and technology adoption explain a positive relationship between attitude and technology adoption [29,45,46]. Based on the previous literature, the hypothesis proposed is:

**Hypothesis 5 (H5)**. *There is a positive relationship between attitude and Fintech adoption in Indonesia.*

### 2.2.6. Financial Literacy (FL)

Financial literacy is the level of personal knowledge in understanding basic financial management information. Financial literacy in this study refers to the ability to understand compound interest, inflation, and risk diversification [47]. Previous studies by Junger and Mietzner [48] and Morgan and Thinh [25] explained that financial literacy positively correlates with Fintech adoption. Liu et al. [49] support the result by adding that financial literacy also aligned with the innovativeness. Based on the above literature, the following hypothesis is proposed for testing financial literacy on Fintech adoption:

**Hypothesis 6a (H6a)**. *There is a positive relationship between financial literacy and Fintech adoption in Indonesia.*

**Hypothesis 6b (H6b)**. *There is a positive indirect relationship between financial literacy and Fintech adoption in Indonesia mediated by user innovativeness.*

### 2.2.7. User Innovativeness (UI)

Lu et al. [50] described user innovativeness as the extent to which a person is willing to experiment with new technology. User open innovation can be accelerated by optimizing the use of external knowledge and information [7]. Meanwhile, Hu et al. [29] describe user innovation as the level of individual acceptance of new products, new technologies, or new services. The readiness to accept the presence of new technology is the main driving factor for technology adoption. The user innovativeness in this research is defined as an intention to try new technologies, to be a pioneer in using the latest technology, and a willingness to experiment with Fintech services. Previous research has explained that user innovativeness has a positive relationship with technology adoption [29,51,52]. Based on the previous literature, the hypothesis proposed is as follows:

**Hypothesis 7 (H7)**. *There is a positive relationship between user innovativeness and Fintech adoption in Indonesia.*

### 2.2.8. Government Support (GS)

Government support plays an essential role in the development of the Fintech industry. The government can be actively involved by making regulations that support the Fintech industry to continue to grow, both for Fintech companies, investors, and service users. The study conducted by Goo and Heo [53] explains that the government's active role has a positive impact on the development of Fintech by reducing uncertainty. Research

by Marakarkandy et al. [54] explains that government support positively correlates with technology adoption. Government support in building hard infrastructure also needs to be supported by users' innovativeness by increasing knowledge about financial and technology literacy, including adopting Fintech products. In this study, government support is associated with the government's role in supporting the Fintech industry, such as laws and regulations that benefit the Fintech industry and infrastructure by expanding the internet network. Align with research framework on Figure 1, the proposed hypothesis is as follows:

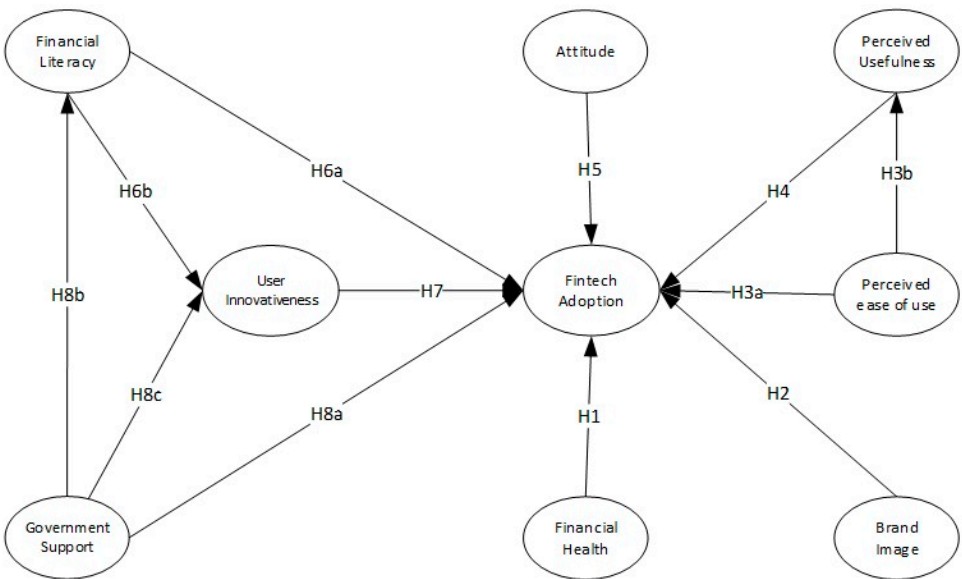

**Figure 1.** Research Framework.

**Hypothesis 8a (H8a)**. *There is a positive relationship between government support and Fintech adoption in Indonesia.*

**Hypothesis 8b (H8b)**. *There is a positive indirect relationship between government support and Fintech adoption in Indonesia mediated by financial literacy.*

**Hypothesis 8c (H8c)**. *There is a positive indirect relationship between government support and Fintech adoption in Indonesia mediated by user innovativeness.*

*2.3. Methodology*

This study uses quantitative research with a Structural Equation Modeling (SEM) approach, using online questionnaire samples with purposive sampling and objects in the study using the criteria of Fintech users in Indonesia. Data were collected online by distributing questionnaires between December 2020 to April 2021, and a pilot survey was conducted first in November 2020 to confirm the validity and accuracy of the questionnaire. Feedbacks from respondents during the pilot survey helped enhance the final survey questionnaire, especially in replacing ambiguous words in the measurement statements with more straightforward, unambiguous, and specific words. After that, questionnaires were distributed to 508 respondents and upon data cleaning, 485 samples were finalized for data analysis.

The questionnaire used a Likert scale of 1 to 5 to measure all research items, from strongly disagree (1) to strongly agree (5). According to Loehlin and Beaujean [55], a good sample size should reflect its population and argued for a minimum sample size to reduce bias in SEM estimation. Hair et al. [56] explains if the population is unknown, the minimum sample number of indicators multiplied by 5; there were 34 indicators in this study, and

thus the minimum sample required for this study is 170 respondents. The 485 respondents in this study exceeded the required minimum sample size.

The SEM with Partial Least Square (PLS)-based approach was applied in this research for testing the research hypotheses. SEM is a data analysis technique that can test a series of relatively complex relationships built simultaneously between the independent and the dependent variable, where each variable can be built from several indicators [56]. SEM aims to estimate the relationship between variables in a model, both between indicators and the relationship between latent variables. SEM is an approach that integrates two analyses, namely factor analysis and regression analysis. For two-stage SEM analysis with PLS, we first test the measurement model in the form of construct validity and reliability of each indicator; and second, test the structural model to determine whether there is influence between variables or correlation between constructs. The structure of the survey and the relevant studies are presented in Table 1.

**Table 1.** Variable description.

| No | Construct Variable | Reference | Indicator | Indicator Code |
|----|---|---|---|---|
| 1 | Fintech Adoption | [54,57] | I will continue using Fintech service | FA1 |
|  |  |  | I haven't used but would like to use Fintech services soon | FA2 |
|  |  |  | I will recommend Fintech services to my friends | FA3 |
| 2 | Financial Health | [25,34] | My earnings are reduced and savings eroded due to pandemic COVID-19 | FH1 |
|  |  |  | Impulsive use of credit card is happening | FH2 |
|  |  |  | Forced to do panic buying and hoarding products | FH3 |
|  |  |  | There is a rise in prices of essential goods | FH4 |
|  |  |  | Tend to withdraw cash more often now | FH5 |
| 3 | Brand Image | [29,37] | I prefer to accept the Fintech services provided by familiar brands | BI1 |
|  |  |  | Fintech overall has a good reputation | BI2 |
|  |  |  | I can recognize Fintech services in Indonesia | BI3 |
| 4 | Perceived Ease to Use | [29,38,39] | It is easy to use Fintech services | PEU1 |
|  |  |  | I think the operation interface of Fintech is friendly and understandable | PEU2 |
|  |  |  | It is easy to have device to use Fintech services (cellphone, APP, WIFI, et al.) | PEU3 |
| 5 | Fintech Perceived Usefulness | [42,43] | Using Fintech can meet my service needs | FPU1 |
|  |  |  | Fintech services can save time | FPU2 |
|  |  |  | Fintech services can improve efficiency | FPU3 |
|  |  |  | Overall, Fintech services are useful to me | FPU4 |
| 6 | Attitude | [29,45,46] | I believe using Fintech services is a good idea | Att1 |
|  |  |  | Using Fintech services gives me pleasant experience | Att2 |
|  |  |  | I am interested in Fintech services | Att3 |
| 7 | Financial Literacy | [47,48] | I have knowledge of compounding interest | FL1 |
|  |  |  | I have knowledge of inflation | FL2 |
|  |  |  | I have knowledge of risk diversification | FL3 |
| 8 | User Innovativeness | [52] | When I hear about a new product, I look for ways to try it | FI1 |
|  |  |  | Among my peers, I am usually the first one to try a new product | FI2 |
|  |  |  | I like to experiment with new Fintech services | FI3 |
| 9 | Government Support | [53,54] | The government support and improve the use of Fintech services | GS1 |
|  |  |  | The government has introduced favorable legislation and regulations for Fintech services | GS2 |
|  |  |  | The government is active in setting up all kinds of infrastructure such as telecom network which has a positive role in promoting Fintech services | GS3 |

## 3. Results

### 3.1. Characteristics of The Respondents

Most respondents in the study were female (62.1%). This is following the characteristics of Fintech users in Indonesia, who are less than 35 years old (72.2%), with the highest education levels being bachelor's degree (42.9%) and high school (36.7%). The majority of respondents (36.6%) in this study (see Table 2) had an income of fewer than 3 million rupiahs per month, and 24.9% have an income of more than 10 million. The majority of respondents in this study (84.3% with 33.6% experiencing once a month, 22.3% two–three times a month, and 28.5% more than four times a month) have used Fintech and 15.7% have never used Fintech; this corresponds to the age characteristics of predominantly young people who better understand and quickly adapt to technology.

**Table 2.** Respondents' characteristics.

| Characteristic | Criteria | Frequency ($n$ = 485) | Percentage (%) |
|---|---|---|---|
| Gender | Male | 184 | 37.9% |
| | Female | 301 | 62.1% |
| Age | 18–25 | 203 | 41.9% |
| | 26–35 | 147 | 30.3% |
| | 36–45 | 124 | 25.6% |
| | >45 | 11 | 2.3% |
| Education | High School | 178 | 36.7% |
| | Bachelor's Degree | 208 | 42.9% |
| | Master's or Doctorate Degree | 99 | 20.4% |
| Net Income | <3 million IDR | 177 | 36.5% |
| | 3–5 million IDR | 96 | 19.8% |
| | 6–10 million IDR | 91 | 18.8% |
| | >10 million IDR | 121 | 24.9% |
| Experience using Fintech | never use | 76 | 15.7% |
| | one time | 163 | 33.6% |
| | 2–3 times | 108 | 22.3% |
| | >4 times | 138 | 28.5% |

### 3.2. Results of the SEM Analysis

Loadings, reliability, convergent validity, and discriminant validity were all considered when evaluating the measurement model. The first step in the evaluation was to look at the loadings of the indicators. Loadings are more significant than 0.7 use for acceptable item reliability. Table 3 shows, all factor loadings have exceeded 0.7. The second step was to determine the reliability of internal consistency [58]. Table 3 shows a composite reliability metric with a threshold value basis of 0.7. The composite reliability values of all constructs were more significant than 0.7, indicating excellent internal consistency. The third step involved determining convergent validity [58]. The extracted average variance (AVE) was used, with a value for each construct that should be greater than 0.5. that the AVE values shown in Table 3 explain for all constructs have exceeded 0.5. The fourth step was to determine the discriminant validity. To ensure that each construct is distinct from the others, we use different metrics. Henseler et al. [59] proposed using the heterotrait-monotrait ratio (HTMT) to test the discriminant validity of constructs. When HTMT values are high, problems arise; for similar constructs, the threshold value is (0.9). Whereas for distinct constructs, it is (0.85). All HTMT value in Table 4 shows a value lower than the threshold.

**Table 3.** Descriptive statistic.

| No | Construct Variable | Indicator | Code | Mean | Std. Deviation |
|---|---|---|---|---|---|
| 1 | Fintech Adoption | I will continue using Fintech service | FA1 | 3.7897 | 0.9432 |
| | | I haven't used but would like to use Fintech services soon | FA2 | 3.1052 | 1.3092 |
| | | I will recommend Fintech services to my friends | FA3 | 3.6825 | 0.9970 |
| 2 | Financial Health | My earnings are reduced and savings eroded due to pandemic COVID-19 | FH1 | 3.6062 | 1.2967 |
| | | Impulsive use of credit card is happening | FH2 | 2.7876 | 1.3735 |
| | | Forced to do panic buying and hoarding products | FH3 | 2.3567 | 1.3231 |
| | | There is a rise in prices of essential goods | FH4 | 3.6309 | 1.1436 |
| | | Tend to withdraw cash more often now | FH5 | 2.9340 | 1.2759 |
| 3 | Brand Image | I prefer to accept the Fintech services provided by familiar brands | BI1 | 4.1381 | 0.9423 |
| | | Fintech overall has a good reputation | BI2 | 3.9237 | 0.8082 |
| | | I can recognize Fintech services in Indonesia | BI3 | 3.8660 | 0.8747 |
| 4 | Perceived Ease to Use | It is easy to use Fintech services | PEU1 | 4.2289 | 0.8401 |
| | | I think the operation interface of Fintech is friendly and understandable | PEU2 | 4.0268 | 0.9345 |
| | | It is easy to have device to use Fintech services (cellphone. APP, WIFI, et al.) | PEU3 | 4.3052 | 0.8167 |
| 5 | Fintech Perceived Usefulness | Using Fintech can meet my service needs | FPU1 | 3.8330 | 1.0138 |
| | | Fintech services can save time | FPU2 | 4.2825 | 0.8934 |
| | | Fintech services can improve efficiency | FPU3 | 4.1299 | 0.9235 |
| | | Overall, Fintech services are useful to me | FPU4 | 4.1691 | 0.9138 |
| 6 | Attitude | I believe using Fintech services is a good idea | Att1 | 3.7753 | 0.8913 |
| | | Using Fintech services gives me pleasant experience | Att2 | 3.8000 | 0.8903 |
| | | I am interested in Fintech services | Att3 | 3.8577 | 0.9284 |
| 7 | Financial Literacy | I have knowledge of compounding interest | FL1 | 3.1464 | 1.2420 |
| | | I have knowledge of inflation | FL2 | 3.6969 | 1.0409 |
| | | I have knowledge of risk diversification | FL3 | 3.3381 | 1.1875 |
| 8 | User Innovativeness | When I hear about a new product, I look for ways to try it | FI1 | 3.4598 | 1.0587 |
| | | Among my peers, I am usually the first one to try a new product | FI2 | 3.0722 | 1.1735 |
| | | I like to experiment with new Fintech services | FI3 | 3.1093 | 1.1599 |
| 9 | Government Support | The government support and improve the use of Fintech services | GS1 | 3.8619 | 0.8403 |
| | | The government has introduced favorable legislation and regulations for Fintech services | GS2 | 3.7649 | 0.8470 |
| | | The government is active in setting up all kinds of infrastructure such as telecom network which has a positive role in promoting Fintech services | GS3 | 3.7320 | 0.9104 |

**Table 4.** Results of the measurement model analysis.

| Construct | Outer Loadings | Cronbrach Alpha's | Composite Reliability | Average Variance Extracted (AVE) |
|---|---|---|---|---|
| Fintech Adoption (Intention) | | 0.8066 | 0.9117 | 0.8378 |
| FA1 | 0.9221 | | | |
| FA3 | 0.9085 | | | |
| Financial Health | | 0.7905 | 0.8756 | 0.7016 |
| FH2 | 0.8797 | | | |
| FH3 | 0.8417 | | | |
| FH5 | 0.7890 | | | |
| Brand Image | | 0.7952 | 0.8797 | 0.7095 |
| BI1 | 0.7848 | | | |
| BI2 | 0.8786 | | | |
| BI3 | 0.8606 | | | |
| Perceived Ease to Use | | 0.8761 | 0.9237 | 0.8015 |
| PEU1 | 0.9085 | | | |
| PEU2 | 0.8814 | | | |
| PEU3 | 0.8956 | | | |
| Fintech Perceived Usefulness | | 0.8697 | 0.9200 | 0.7931 |
| FPU1 | 0.8774 | | | |
| FPU3 | 0.8963 | | | |
| FPU4 | 0.8978 | | | |
| Attitude | | 1 | 1 | 1 |
| Att1 | 1 | | | |
| Financial Literacy | | 0.8279 | 0.8970 | 0.7440 |
| FL1 | 0.8800 | | | |
| FL2 | 0.8317 | | | |
| FL3 | 0.8751 | | | |
| User Innovativeness | | 0.8728 | 0.9218 | 0.7972 |
| FI1 | 0.8725 | | | |
| FI2 | 0.9097 | | | |
| FI3 | 0.8960 | | | |
| Government Support | | 0.8566 | 0.9128 | 0.7774 |
| GS1 | 0.8815 | | | |
| GS2 | 0.9126 | | | |
| GS3 | 0.8499 | | | |

Note: because Outer loadings factor under 0.7 "FA2, FH1, FH4, Att2, Att3, FPU3" exclude form indicator.

When the measurement model assessment is satisfactory, Estimate the structural model by examining its explanatory power and the statistical significance of the path coefficient. Before evaluating the structural model, the multicollinearity of constructs will review. Collinearity issues arise when the variance inflation factor (VIF) exceeds 5 [58]. Table 5 shows that the VIF values for all construct items were less than 5. the model's explanatory power assesses use coefficient of determination ($R^2$). The coefficient of determination calculates for all endogenous constructs. The $R^2$ value for the usage intention construct, as shown in Table 6, indicates that the model has moderate explanatory power.

**Table 5.** Discriminant validity heterotrait-monotrait (HTMT) values.

|  | AT | BI | FA | FH | FL | PEU | FPU | GS |
|---|---|---|---|---|---|---|---|---|
| AT |  |  |  |  |  |  |  |  |
| BI | 0.678 |  |  |  |  |  |  |  |
| FI | 0.822 | 0.839 |  |  |  |  |  |  |
| FH | 0.224 | 0.170 | 0.271 |  |  |  |  |  |
| FL | 0.371 | 0.444 | 0.506 | 0.206 |  |  |  |  |
| PEU | 0.563 | 0.857 | 0.703 | 0.090 | 0.410 |  |  |  |
| FPU | 0.704 | 0.794 | 0.834 | 0.085 | 0.453 | 0.866 |  |  |
| GS | 0.610 | 0.826 | 0.716 | 0.323 | 0.393 | 0.646 | 0.638 |  |
| US | 0.587 | 0.564 | 0.669 | 0.513 | 0.461 | 0.399 | 0.411 | 0.623 |

Note: The meaning of "AT = attitude; BI = Brand Image; FA = Fintech Adoption, FH = Financial Health; FL= Financial Literacy; PEU = Perceived Ease to Use; FPU = Fintech Perceived Usefulness; GS = Government Support".

**Table 6.** Variance inflation (VIF) values.

| Construct | VIF | Construct | VIF |
|---|---|---|---|
| Fintech Adoption (Intention) |  | Attitude |  |
| FA1 | 1.84102519 | Att1 | 1 |
| FA3 | 1.84102519 |  |  |
| Financial Health |  | Financial Literacy |  |
| FH2 | 1.69585048 | FL1 | 2.03073350 |
| FH3 | 1.84267147 | FL2 | 1.67142607 |
| FH5 | 1.54717257 | FL3 | 2.15740800 |
| Brand Image |  | User Innovativeness |  |
| BI1 | 1.52589600 | FI1 | 2.01717081 |
| BI2 | 1.92993997 | FI2 | 2.90770523 |
| BI3 | 1.77368794 | FI3 | 2.52508112 |
| Perceived Ease to Use |  | Government Support |  |
| PEU1 | 2.57243586 | GS1 | 2.17803232 |
| PEU2 | 2.20288289 | GS2 | 2.60156244 |
| PEU3 | 2.43797089 | GS3 | 1.9426931 |
| Fintech Perceived Usefulness |  |  |  |
| FPU1 | 2.13003015 |  |  |
| FPU3 | 2.50413543 |  |  |
| FPU4 | 2.32382409 |  |  |

The $R^2$ in Table 7 shows the ability of the research model to explain the contribution of the determinants to explain changes that occur in Fintech adoption, and $R^2$ can also assess how well the model is expected to explain and predict future outcomes. Thus, a high $R^2$ value can increase the probability of correct predictions [58]. This research model explains a substantial variant of Fintech adoption ($R^2$= 0.687, Table 7), which means the brand image, Fintech perceived usefulness, user attitude, financial literacy, and user innovativeness explain 68.7% of the variation in Fintech adoption. In addition, the research model explains user innovativeness as a mediator for government support and financial literacy in Fintech adoption ($R^2$ = 0.345), which shows that the user innovativeness accounts for 34.5% of the variation in Fintech adoption.

**Table 7.** Coefficient of determination ($R^2$) values.

|  | R-Square | R-Square Adjusted |
| --- | --- | --- |
| Fintech Adoption | 0.687 | 0.682 |
| Financial Literacy | 0.110 | 0.108 |
| Fintech Perceived Usefulness | 0.575 | 0.574 |
| User Innovativeness | 0.345 | 0.342 |

A complete boot-strapping procedure with 5000 samples uses to test the statistical significance of the path coefficients. Figure 2 depicts the findings of the structural model analysis.

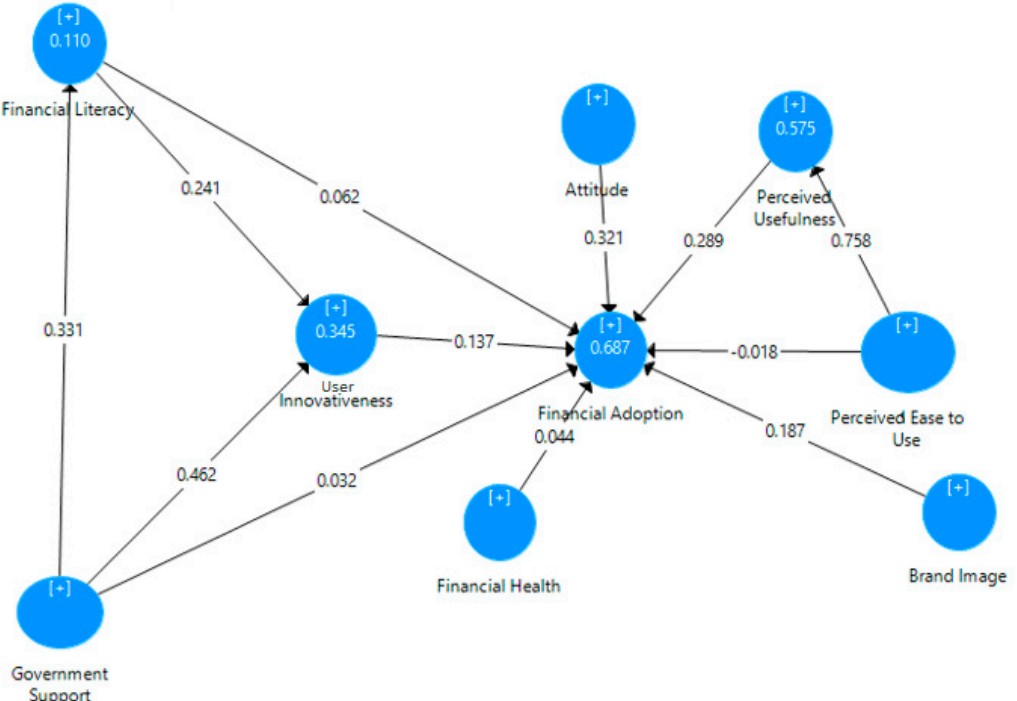

**Figure 2.** Results of hypothesis tests.

As shown in Table 8 shows eight direct effect hypotheses and four indirect effect hypotheses. Of the eight direct effect hypotheses, five hypotheses are supported, except H1, H3a, and H8a. Brand image (H2; β = 0.19), perceived usefulness (H4; β = 0.29), attitude (H5; β = 0.32), and user innovativeness (H7; β = 0.14) have a positive and significant direct effect to Fintech adoption; thus H2, H4, H5, H6a, and H7 are supported. On the other hand, financial health, Fintech ease to use, and government support have a non-significant direct effect on Fintech adoption.

Meanwhile, Table 8 shows all indirect effects hypotheses are supported. According to these results, the indirect effect between perceived ease to use (H3b; β = 0.22) and Fintech adoption is statistically significant, although the direct effect of Fintech ease of use to Fintech adoption is not significant. This indicates that perceived usefulness fully mediates the relationship between perceived ease to use and Fintech adoption. Furthermore, government support (H8b; β = 0.02) has a positive significant indirect effect on Fintech adoption mediated by financial literacy. Also, government support (H8c; β = 0.06) has a positive significant indirect effect on Fintech adoption mediated by user innovativeness, although the direct effect of government support on Fintech adoption is not significant. This indicates that financial literacy and user innovativeness fully mediate the relationship between government support and Fintech adoption. Meanwhile, financial literacy has an indirect effect on Fintech adoption, with user innovativeness becoming a mediating variable.

**Table 8.** Hypothesis Testing.

| | Hypotheses | Original Sample/β | *p*-Value | Decision |
|---|---|---|---|---|
| H1 | Financial Health →Fintech Adoption | 0.0438 | 0.1355 | not supported |
| H2 | Brand Image →Fintech Adoption | 0.1872 | 0.0002 *** | supported |
| H3(a) | Perceived Ease to Use →Fintech Adoption | −0.0179 | 0.7248 | not supported |
| H3(b) | Perceived Ease to Use →Fintech Perceived Usefulness →Fintech Adoption | 0.2190 | 0 *** | supported |
| H4 | Fintech Perceived Usefulness →Fintech Adoption | 0.2889 | 0 *** | supported |
| H5 | Attitude →Fintech Adoption | 0.3206 | 0 *** | supported |
| H6(a) | Financial Literacy →Fintech Adoption | 0.0624 | 0.0318 ** | supported |
| H6(b) | Financial Literacy →User Innovativeness →Fintech Adoption | 0.0329 | 0.0018 *** | supported |
| H7 | User Innovativeness →Fintech Adoption | 0.1369 | 0.0001 *** | supported |
| H8(a) | Government Support →Fintech Adoption | 0.0317 | 0.4797 | not supported |
| H8(b) | Government Support →Financial Literacy →Fintech Adoption | 0.021 | 0.0494 ** | supported |
| H8(c) | Government Support →User Innovativeness →Fintech Adoption | 0.0632 | 0.0003 *** | supported |

Note: ** and *** represent significance at 5% and 1% respectively and the t-values are given in parentheses.

## 4. Discussion

### 4.1. Fintech Adoption in Indonesia

The purpose of this study was to empirically measure user innovativeness, attitude, brand image and financial health toward Fintech adoption during the COVID-19 crisis. To achieve this goal, social constructs from the extended hybrid framework, such as internal issues (user innovativeness, attitude, financial literacy, and financial health) and external issues (government support and brand image), were integrated into the TAM model.

Descriptive findings explain the characteristics of Indonesian society where young people under 35 years old, often referred to as millennials, dominate the experience of using Fintech. This is in line with research from Association of Indonesian Internet Service Providers [60] that explains Indonesia is experiencing relatively rapid development, with internet users reaching 73.7% of the total population of Indonesia. Fintech is considered an alternative or solution that can facilitate Indonesian people in conducting their financial transactions, especially the role of Fintech in payment transactions. The Fintech ecosystem in Indonesia is strengthened by regulation and innovation to improve service infrastructure in the field of technology. Pandemic accelerates this by forcing or pressure people who cannot interact directly must switch indirectly that is utilizing the role of technology in economic activities.

The study findings reveal that user attitude has the most significant direct impact on individuals' intention to use Fintech during crisis. These results are consistent with the results from previous studies [29,46,48]. Besides that, brand image [29,37], perceived usefulness [41,61,62], financial literacy [25,47,48], and user innovativeness [29,51,52,63] had a significant direct impact on Fintech user adoption. Surprisingly, financial health, perceived ease to use and government support did not have a direct impact on Fintech adoption, although Fintech easy to use, financial health and government support do not have a direct effect. Our findings show government support and perceived ease to

use has an indirect impact on Fintech use adoption. These findings can be explained as suggesting the external factor such as government support does not have a direct effect on Fintech user adoption. Still, government and policy makers can make policies that improve the community's financial literacy and user innovativeness to increase the level of Fintech user adoption in the community, especially entrepreneurs and Micro Small Medium Enterprises (MSMEs).

Based on our findings, internal factors such as financial health do not hinder the public in using financial technology. However, in the period of the COVID-19 pandemic, the discussion revealed many people and entrepreneurs have problems in financial health. Although, direct subsidies from the government are the same as the lowest living costs make the community and entrepreneurs survive. Another finding said external factors such as brand image have contributed to elevated financial technology user adoption. Therefore, to improve the brand image of Fintech services, companies should prioritize convenience and security as an integral part of the Fintech service to encourage users' Fintech services adoption.

*4.2. User Innovativeness on Fintech Adoption*

Rogers [12] describes individual innovation as a person's readiness to adapt relatively faster than others, in this study with regards to Fintech adoption. Individual readiness to innovation encourage user open innovation and it correlates with openness to new experiences, creativity, leadership, and a courageous attitude in taking risks [7,64]. Previously, individual innovativeness theory was commonly applied in education research, such as to measure individual readiness to adapt to digital learning process. Innovation among people is viewed as a multi-factorial construct. Pratoom and Savatsomboon [65] describe the factors that impact individual innovation in 1526 member groups in Thailand and found creativity, self-leadership, and knowledge management having influence towards individual innovation. Recent studies by Aldahdouh et al. [66] from 315 employees at Tampere University, Finland, depicts a person's psychological factor as the main trigger for individual innovativeness. Recent studies that integrate user innovativeness with technology adoption in the financial sector were conducted by Yoon and Lim [14]. In this study, user innovativeness is integrated with the individual's willingness to find ways to operate new technology, the speed at which a person tries new technology compared to others, including the level of individual preference in conducting experiments on Fintech services as prescribed by Hu et al. [29]. Corroborating with recent findings, our study also finds that user innovativeness does significantly influence users' Fintech adoption.

This study finds that user innovation has an essential role in Fintech adoption in Indonesia. User innovation has a direct and indirect impact on user acceptance of Fintech services. The direct impact of user innovation on Fintech adoption is shown by the significant relationship between the two variables. Meanwhile, the indirect effect is shown through the role of user innovation as mediating factor for financial literacy and government support for Fintech adoption. The result is undoubtedly surprising since user innovation could indirectly benefit factors associated to users' internal knowledge of financial literacy and the external factor reflected in government support. This suggests that internal and external factors play an essential role in driving people's innovation to adapt to new technology, particularly for Fintech adoption.

## 5. Conclusions, Implications, Limitations, and Recommendations

*5.1. Conclusions*

This study is focused on Fintech adoption in Indonesia. Fintech activities in this country have developed very rapidly, both Fintech services pioneered by incumbent companies in the financial and banking industry and internet-only bank start-ups that optimize internet-based information technology as a business model. To investigate the Fintech adoption, various important technology acceptance factors are identified based on the

literature, evaluated from the information that has been collected through questionnaires among Fintech users in Indonesia.

The findings crucial factors that drive Fintech adoption for Indonesian users. The variable attitude was the most significant determining factor, and on the contrary users' financial literacy contributed the least to Fintech adoption. The research findings also show that user innovativeness impacts, directly and indirectly Fintech adoption in Indonesia. It is necessary to empower the regulation and educate society since government support affect Fintech adoption through user innovativeness. Public access to the financial sector through digitization is an essential initiative for local governments to drive equality and welfare; and promote United Nation's SDG number 8, with regards access to financial services for everyone. Even though this study found that financial health does not affect the Fintech adoption, it is essential for society to embrace Fintech services. More importantly during the COVID-19 pandemic, where the interaction between human is limited.

### 5.2. Implications

Fintech adoption needs to be optimized through macro and micro levels. The government participates in the macro level by building infrastructure such as an internet network to provide access for users of financial services both in the rural and urban areas. The active involvement of financial industry players by providing financial products according to the community's needs is also an essential element in expanding access to formal financial services. On the micro-level, user open innovation needs to be accelerated by optimizing the use of external information to adopt the new technology [7]. Optimizing the involvement of all related parties such as governments, academics, researchers, and other stakeholders, plays a pivotal role in increasing financial literacy and inclusion index in a country.

### 5.3. Limitations and Recommendations

This study collected data from 458 respondents, then analyzed it to investigate the Fintech adoption in Indonesia. Currently, the respondents are only Indonesian, so the Fintech adoption with respondent from other countries could produce different result from this study. Despite the sample size of this study is relatively larger than previous research on Fintech adoption in Indonesia, the investigation only focuses on the individual level. Broader studies can be conducted with different samples, such as MSMEs, whose contribution is significant to the Indonesian economy. In addition, the adoption of Fintech that focuses on women-MSMEs also needs observation, in line with the UNSDGs number five target to promote the achievement of gender equality. The data collection process during a pandemic COVID-19 is also a challenge because researchers are unable to properly understand respondent's answers.

Future research may modify the theoretical framework of this study for other countries. This research can also be expanded by adding new dimensions such as cultural factors and the geographic location of respondents. In addition, research also needs to focus more specifically on certain Fintech services.

**Author Contributions:** Conceptualization, B.S. and R.J.N.; methodology, D.P.N. and R.J.N.; software, D.P.N.; validation, B.S., D.P.N., A.I. and R.J.N.; formal analysis, D.P.N. and R.J.N.; investigation, B.S., D.P.N. and A.I.; resources, B.S.; data curation, B.S. and D.P.N.; writing—original draft preparation, B.S., D.P.N. and A.I.; writing—review and editing, B.S., D.P.N., A.I. and R.J.N.; supervision, R.J.N. and Z.Z.; project administration, B.S., R.J.N., and Z.Z.; funding acquisition, B.S., and Z.Z. All authors have read and agreed to the published version of the manuscript.

**Funding:** This research received no external funding.

**Institutional Review Board Statement:** Not applicable.

**Informed Consent Statement:** Informed consent was obtained from all subjects involved in the study.

**Data Availability Statement:** The data presented in this study are available on request from the corresponding author.

**Acknowledgments:** We would like to thank, first, the reviewers who gave us suggestions on how to develop this article. Secondly, to the respondents in Indonesia who are willing to complete the questionnaire.

**Conflicts of Interest:** The authors declare no conflict of interest.

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
