# Peer review of "User Innovativeness and Fintech Adoption in Indonesia"

_2199-8531, doi:10.3390/joitmc7030188_

Round 1

Reviewer 1 Report

the authors should improve the results comment connecting the results with the literature review section. Moreover, I think that the hypothesis are too much and should be reduced.

Author Response

Dear Reviewer,

Thank you very much for your suggestions and we have revised the manuscript as suggested. We look forward to hearing back from you at your earliest convenience.

Best regards,

Authors

Reviewer 2 Report

Regarding the choice of their topic, the authors indicate a growing area which has implications for practice as well as for theory. In this sense, the approach not only fits squarely within the scope of JoI:TMC, but it is very likely that it will pass the test of time.

Nevertheless, for such a new and emerging topic to be analyzed, more emphasis is required on previous work. For example, I did not see some of the most recent and related work in their literature review:

https://doi.org/10.1145/3400934.3400939

https://doi.org/10.3390/joitmc7010088 

This can be one possible area of improvement. 

In addition, more emphasis must be given to the "value added" of the paper - this is surely not restricted for those studying Indonesia or policymakers in Indonesia. What are the implications that we can generalize? In addition, what are the theoretical/conceptual implications regarding the market shaping power of FinTech and its adoption process in consumer-producer dynamics, especially with regard to co-development?

For the latter point, the authors may want to consult

Author Response

(The authors gave the same response as above.)

Reviewer 3 Report

Please refer to attached opinion

Author Response

(The authors gave the same response as above.)
